# CIL Gold Loss Characterization within Oxidized Leach Tails: Creating a Synergistic Approach between Mineralogical Characterization, Diagnostic Leach Tests, and Preg-Robbing Tests

**Mohamed Edahbi [1,2,\*], Raphaël Mermillod-Blondin [3], Benoît Plante [1] and Mostafa Benzaazoua [1]**

[1] Institut de Recherche en Mines et Environnement (IRME), Université du Québec en Abitibi-Témiscamingue (UQAT), 445 boul de l'Université, Rouyn-Noranda, QC J9X 5E4, Canada; benoit.plante@uqat.ca (B.P.); mostafa.Benzaazoua@uqat.ca (M.B.)

[2] Department of Geological Sciences, University of Saskatchewan, Saskatoon, SK S7N 5E2, Canada

[3] Agnico-Eagle Mines, 10 200, Route de Preissac, Rouyn-Noranda, QC J0Y 1C0, Canada; raphael.mermillod-blondin@agnicoeagle.com

\* Correspondence: Mohamed.Edahbi@uqat.ca; Tel.: +1-306-966-5720 (ext. 5720)

**Abstract:** A double refractory gold ore contains gold particles locked in sulphides, solid-solution in arsenopyrite, and preg-robbing material such as carbonaceous matter, and so on. The diagnostic leach test (DLT) and preg-robbing (PR) approaches are widely used to investigate the occurrence and the distribution of refractory gold. DLT serves to qualitatively evaluate the gold occurrences within the ore. Preg-robbing, or the ore's capacity to fix dissolved gold, is evaluated to determine physical surface interactions (preg-borrowing) and chemical interactions (preg-robbing). The objective of this project is to characterize the refractory gold in Agnico Eagle Mine's Kittilä ore using the DLT and PRT approaches coupled with mineralogical analyses to confirm testing. The studied material was sampled from the metallurgical circuit following carbon in leach (CIL) treatment at the outlet of the autoclave in order to investigate the effect of the autoclave treatment on the occurrence and distribution of gold. Different reagents were used in the DLT procedure: sodium carbonate ($Na_2CO_3$), sodium hydroxide (NaOH), hydrochloric acid (HCl), and nitric acid ($HNO_3$). The final residue was roasted at a temperature of around 900 °C. These reagents were selected based on the mineralogical composition of the studied samples. After each leaching test/roasting, cyanide leaching with activated carbon was required to recover gold cyanide. The results show that gold is present in two forms (native and/or refractory): to a small extent in its native form and in its refractory form as association with sulfide minerals (i.e., arsenopyrite and pyrite) and autoclave secondary minerals that have been produced during the oxidation and neutralization processes such as iron oxides, iron sulfates, and calcium sulfate (i.e., hematite and jarosite), along with carbonaceous matter. The results of DLT indicate that 25–35% of the gold in the tails is nonrecoverable, as it is locked in silicates, and 20–40% is autoclave products. A regrind can help to mitigate the gold losses by liberating the Au-bearing sulphide minerals encapsulated within silicates.

**Keywords:** refractory gold; characterization; diagnostic leaching tests; preg-robbing tests; cyanide leaching

## 1. Introduction

The gold from the Kittilä mine is in a refractory form. The refractory gold is also called submicroscopic gold, which specifically refers to atomic gold substituted into the matrix of other

minerals such as arsenopyrite and pyrite [1]. Only 10–20% of the gold is in its native form and the remaining fraction is disseminated within its Au-bearing minerals. It is also reported that refractory gold-bearing sulphides are commonly associated with organic carbon (i.e., amorphous and graphite) [2–5]. Moreover, the gold ore from the Kittilä mine is also associated with carbonaceous matter, which could reduce the gold recovery owing to its sorption potential (preg-robbing phenomena) [6]. Before the cyanidation process, the refractory gold must be oxidized under pressure oxidation and/or bioleached to facilitate its liberation [7,8]. Moreover, the physical (e.g., particle size and specific surface area) and mineralogical (e.g., degree of liberation) characterization of the organic carbon could affect the extraction of the refractory gold [9]. In the Kittilä case, the gold loss associated to carbon prefloat before the autoclave treatment is low because of the lack of the preg-robbing. Consequently, any gold losses during flotation would be the result of gold association in the original ore with the prefloat tails.

The metallurgical treatment of gold is generally performed, to recover gold from primary and secondary minerals, in two stages [10]: gold extraction and refining of the metal. The extraction process is carried out using cyanidation [11,12]. The gold is recovered by activated carbon. However, the extraction of refractory gold is generally difficult to accomplish because of encapsulated/locked gold within gangue minerals, solid-solution gold in pyrite and arsenopyrite [13], and cyanide consumption due to high amounts of sulfide minerals [14–16]. Consequently, the performance of reagents is highly limited, despite the use of various reagents at different conditions (i.e., acidic and/or basic) [17–19]. Pre-treatments are required to improve gold recovery at economically viable levels [20,21]. Moreover, the beneficiation techniques such as gravity separation, flotation, roasting, high pressure oxidation, and biological oxidation are the most common techniques used in the gold industry to extract refractory gold [15,22–24]. A gold concentrate can be produced through a combination of metallurgical techniques such as the hydrometallurgical process, followed by pyrometallurgical treatment [14]. Moreover, the gold might be recovered as a by-product from base metals industries (i.e., copper and zinc smelters, nickel sulfide) using various processes such as the treatment of gravity concentrates or smelting of metal concentrates (i.e., Cu, Zn, Ni, and Pb) [25]. However, all these methods are costly and require a major long-term investment. Mineralogical and surface characterization techniques (inductively coupled plasma ICP-mass spectrometry MS, secondary ion MS SIMS, and electron probe micro-analyzer EPMA) are recommended to improve the metallurgical efficiency as they allow the refractory gold-bearing minerals and their degree of liberation to be determined [26,27].

Diagnostic leach tests (DLT) and preg-robbing tests (PRT) are widely used to investigate the occurrence and distribution of refractory gold [28–34]. DLT is a semi-quantitative procedure used to quantify and evaluate the deportment/occurrences of gold in various Au-bearing minerals within the ore. The PRT indicates the ore's capacity to fix dissolved gold. It evaluates the physical surface interactions (preg-borrowing) and chemical interactions (preg-robbing). Gold recovery at the Kittilä mine does not depend on the gold grade (Figure 1A), but especially on the content and the maturity of the carbonaceous matter (Figure 1B–D). As demonstrated in Figure 1C,D, a high carbon content in the tailing leads to high gold losses. Similar observations were found in the literature, particularly in Helm's studies, where the carbon maturity was investigated using Raman spectroscopy [35,36]. In the present study, the estimation of the ore's sorption capacity was initially evaluated by cyanidation tests using activated carbon. Recently, high gold losses were observed. Furthermore, the ore mineralogy of the feed samples is refractory. The pressure oxidation leads to complete alteration of the mineralogy, rendering the feed sample unsuitable for examination. For this reason, the present study mainly focuses on the tailings samples to infer the mechanisms responsible for the gold losses and to propose particular recommendations during the gold recovery. The typical relationship between the total organic carbon (TOC) content and the sorption capacity was not observed, which suggests that the carbon preg-robbing capacities may vary within the deposit. Many cases can be found in the literature where preg-robbing is significantly different in ore taken from the footwall compared with that from the hanging wall side of the deposit, owing to differences in the extent of alteration of the carbonaceous matter [37]. The present study aims to test this hypothesis.

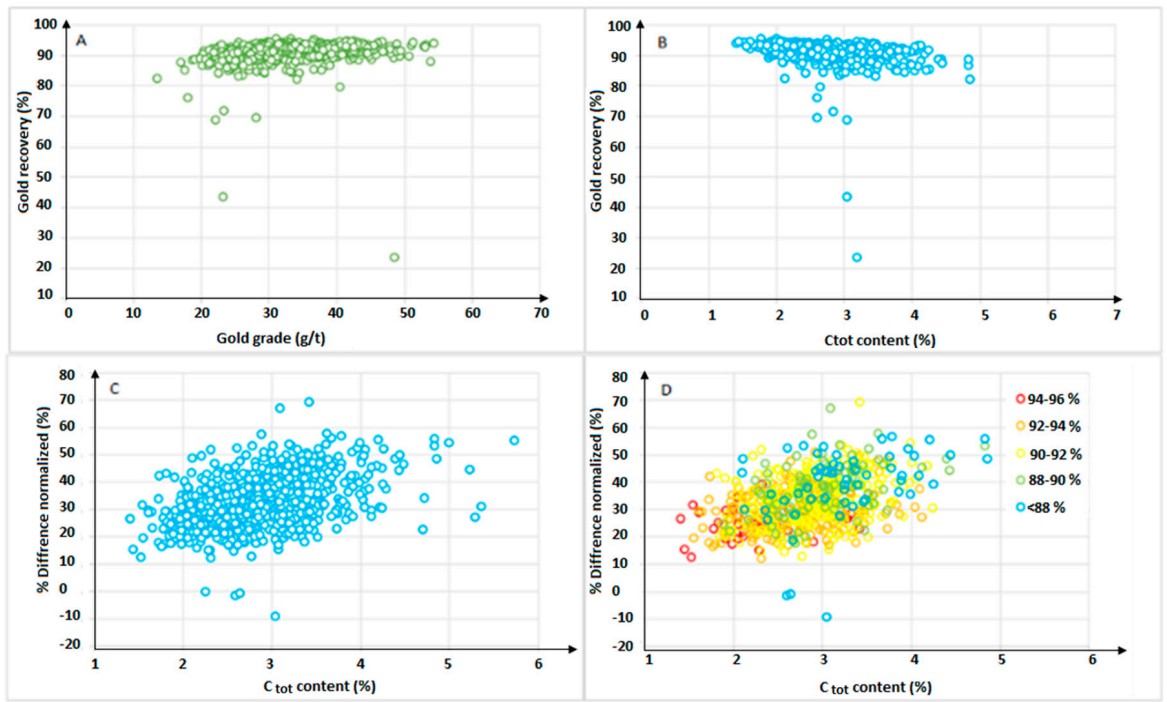

**Figure 1.** Gold recovery and % difference normalized as a function of Au grade (**A,B**) and $C_{tot}$ content (**C,D**), respectively. The % difference = 1 − (recovery with activated carbon/recovery without activated carbon) (Kittilä mine 2017, unpublished report).

Currently, further research is necessary to fill in important gaps regarding refractory gold recovery. In the case of the Kittilä mine, it is observed that the gold recovery depends the TOC content and its reactivity (calculation via bottle roll tests with/without activated carbon). The challenges are to determine the link between the gold in the tailings and the total TOC, and the relationship between reactivity and carbon maturity. The specific aims of this study are as follows: (1) to characterize carbon in leach (CIL) gold loss within pressure oxidation (POX) leach tails, (2) to evaluate the gold distribution in four different samples by DLT in alkaline and acid conditions, and (3) to estimate the PR capacity of the carbonaceous materials using different adsorption tests (with and without activated carbon). The present research aimed at achieving these objectives by adopting an original methodology aiming to explore, for the first time, this aspect using a synergistic approach between mineralogical characterization, diagnostic leach tests, and preg-robbing tests.

## 2. Material and Methods

Mineralogical characterization, DLT, and adsorption tests were performed to study the gold distribution and preg-robbing, using four samples taken from the mill carbon in leach (CIL) tailings at the Kittilä Mine in Finland (oxidized tailings). The following section describes the materials and methods used in more detail.

### 2.1. Sampling

Approximately 10 kg of four CIL tailings (identified as Tail-17, Tail-19, Tail-22, and Tail-25, taken September 17, 19, 22, and 25, respectively, in 2016) were sampled from the metallurgical circuit following CIL treatment (Figure 2). These samples were dried, de-agglomerated, and homogenized separately.

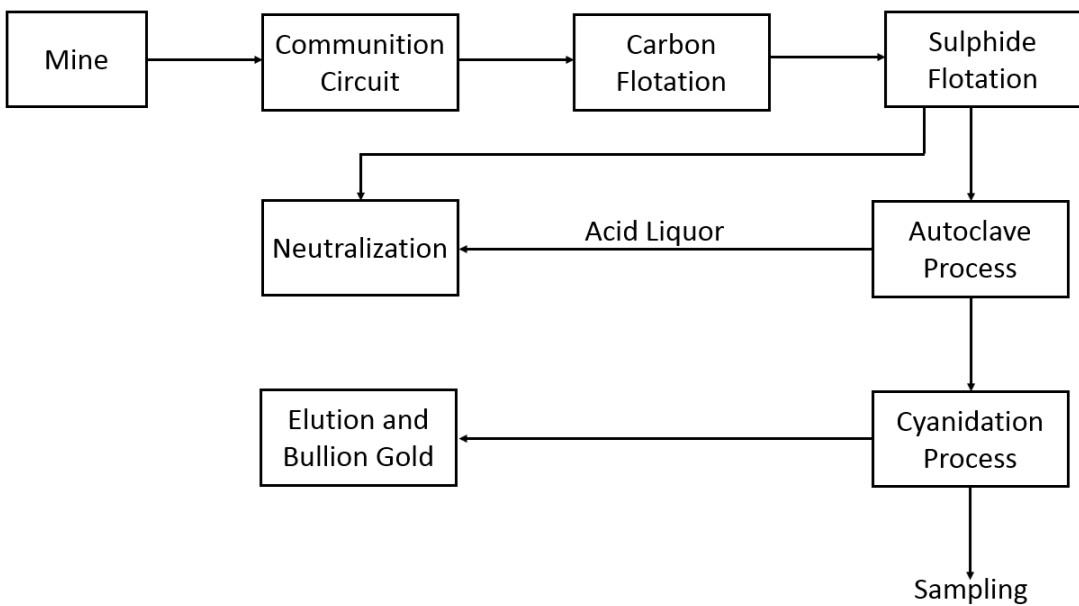

**Figure 2.** Au processing flow chart for Kittilä Mine and sampling locations.

*2.2. Analytical Methods*

The grain size distributions were analyzed using a laser analyzer (Malvern Mastersizer S). The chemical compositions of the samples were analyzed by inductively coupled plasma atomic emission spectroscopy (ICP-AES) after a multi-acid digestion (HCl, $HNO_3$, $Br_2$, and HF) of a 500 mg aliquot. Leachates from testing were analyzed with ICP-AES following acidification of a filtered (0.45 μm) aliquot to about 2% $HNO_3$. The S and C content were determined using an ELTRA CS-2000 induction furnace.

The mineralogical study was performed on polished sections of the four samples by optical microscope (OM—Zeiss Axio imager.M2m, Zeiss, QC, Canada) and scanning electron microscope (SEM—Hitachi S-3500N, Hitachi, QC, Canada), equipped with a microanalysis system (energy dispersive spectroscopy (EDS) INCA XMax 20 mm$^2$ SDD, Oxford, QC, Canada). The OM observations mainly focused on sulfide minerals, while the analyses of other minerals were done using the SEM-EDS system. During sulfide oxidation in autoclave, hematite and jarosite are formed and, consequently, gold could be occluded/adsorbed by the secondary minerals. Moreover, it is interesting to note that jarosite could be also formed during the neutralization processes. During the autoclave process, the time residence for a long term (>2 h) and in high temperature (>200 °C) are well known to favor the precipitation of the secondary minerals that trap gold. The optimization of these two parameters could help to avoid the gold loss in the autoclave. Moreover, the presence of sulfates and iron coming from the sulphide oxidation and water process (autoclave feed water) with optimal conditions of pH and Eh has allowed the formation of these secondary minerals (e.g., jarosite and hematite).

The crystalline minerals were identified by X-ray diffraction (XRD; Brucker D8 Advance, Brucker Ltd, QC, Canada, with a detection limit and precision of approximately 0.1% to 0.5%, operating with a copper cathode, Kα radiation) using the DIFFRACT.EVA program (Version 3.1, Bruker, Milton, ON, Canada), and quantified using the TOPAS 4.2 program (Version 4.2, Bruker, Milton, ON, Canada) based on a Rietveld (1993) interpretation.

The dynamic secondary ion mass spectrometry (D-SIMS) technique is a benchmark technique for analyzing submicroscopic (invisible) gold in minerals [38,39]. The submicroscopic gold detected and quantified by the D-SIMS instrument is refractory gold, that is, it is locked within the crystalline structure of the mineral phase (most often in sulfide minerals) and cannot be directly released by the cyanide leach process. This type of gold may be present as finely disseminated, colloidal-size gold particles (<0.5 μm) or as a solid solution within the sulfide mineral matrix. During the D-SIMS analysis,

an ion beam removes consecutive layers of material from the surface of the polished mineral grains and generates depth profiles of the distribution of the chosen elements of interest. Examples of D-SIMS depth profiles show the distribution of the basic matrix elements (S, Fe) as well as the trace elements (Au and As). The spikes in the gold signal intensity in the depth profiles represent colloidal gold, and the yellow-coloured areas represent the approximate size of this colloidal-type, submicroscopic gold. The typical size is in the range of 100–200 nm. Dynamic SIMS depth profiles for solid solution, submicroscopic gold show a steady (flat) Au signal similar to the matrix elements, but with different levels of intensity depending on the concentration of submicroscopic gold present in the mineral phase.

The marked mineral particles of interest were analyzed using the Cameca IMS-3f SIMS instrument (Cameca, ON, Canada), and concentration depth profiles for Au, As, S, and Fe were produced. The quantification of the gold and arsenic in pyrite, hematite, and jarosite was established using internal mineral specific standards.

### 2.3. Diagnostic Leach Tests

Selective alkaline reagents (NaCO$_3$, NaOH), acid reagents (HCl, and HNO$_3$), and roasting were used to evaluate the gold distribution within samples during gold DLTs. To improve the dissolution of gold-bearing minerals, all pulps (50% solids by weight) were heated to temperatures of 80 °C. After each oxidation test, cyanide leaching with activated carbon was used to extract the dissolved gold. The methodology followed for the diagnostic leach procedure is described in Figure 3, as developed by Kalonji Kabambi (2015) [40]. Five stages can be distinguished:

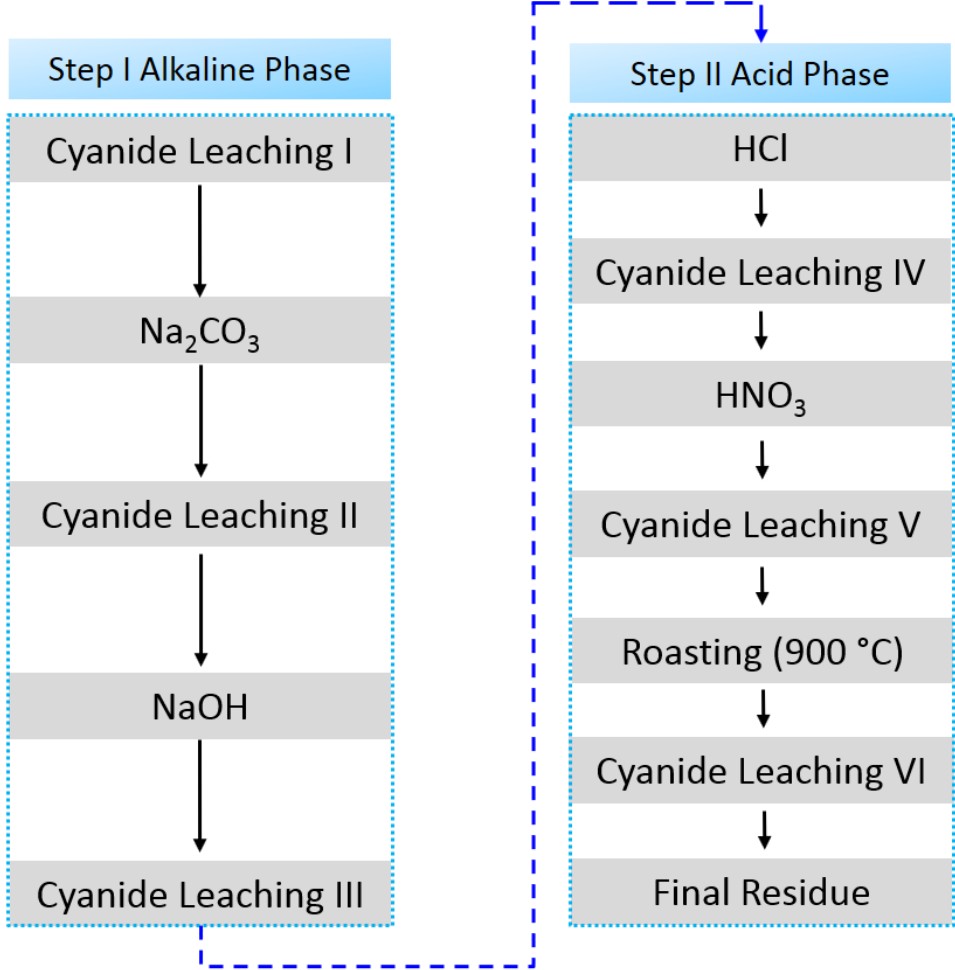

**Figure 3.** Diagnostic leach tests for refractory gold ores.

Stage 1: Leach sample reacts with sodium carbonate to dissolve labile gypsum and free gold;
Stage 2: Leach sample reacts with sodium hydroxide to dissolve labile jarosite;
Stage 3: Leach sample reacts with hydrochloric acid to dissolve iron oxide, especially hematite;
Stage 4: Leach sample reacts with nitric acid to dissolve sulfide minerals, especially pyrite.
Roasting at 900 °C to burn native carbon into volatile $CO_2$.

After each stage, the material is cyanide loaded in a tank for 16 hours with activated carbon (500 g/t), 5 g/t of NaCN, 50 g/t of carbon, 0.3 g/t of $Pb(NO_3)$ to mitigate passivation by the remaining sulphides, 8 ppm of $O_2$, and 50% of solid.

## 2.4. Preg-Robbing Tests

The preg-robbing tests allow to determine the preg-robbing potential of the sample by following the evolution of a known amount of gold in cyanide solution during its contact with the sample. Then, the capacity sorption of the sample is determined by re-measuring the residual concentration of gold after a residence time. To determine the preg-robbing value with more accuracy, the material must be characterized before starting the test. The method is based on a comparative procedure using five parallel cyanide leaching tests. The details of the conditions are as follows:

Normal Test (N): Leaching with only a cyanide solution;

Gold Test (G): Leaching with a standard solution of gold cyanide;

Normal Carbon Test (NC): Leaching with a solution of cyanide in contact with the material and activated carbon;

Gold Carbon Test (GC): Leaching with a standard solution of gold cyanide in contact with the material and activated carbon (50 g/t);

Carbon Test (C): Contact of a standard gold cyanide solution with the activated carbon, without material.

The concentration of standard gold cyanide used in this study was determined through a series of kinetic tests. The objective was to determine which concentration to use without saturating the adsorption sites of the carbonaceous matter within the ore.

The standard gold solution concentration is around 100 mg/L, obtained with $KAu(CN)_2$ salt. At this concentration, the 50, 100, 200, 400, 600, and 800 mL of Au standard correspond to 5, 10, 20, 40, 60, and 80 mg/L, respectively.

## 3. Results and Discussion

### 3.1. Physical and Chemical Analysis

The particle size distribution is of great importance in the assessment of diagnostic leach and preg-robbing tests. Fine particles are known to be more reactive than coarse particles. The particle size distributions of the material are shown in Figure 4. The samples were de-agglomerated to obtain a similar particle size distribution in order to ensure equivalent experimental conditions. The $D_{50}$ and $D_{90}$ values varied between 7–7.5 μm and 40–45 μm, respectively, for all samples.

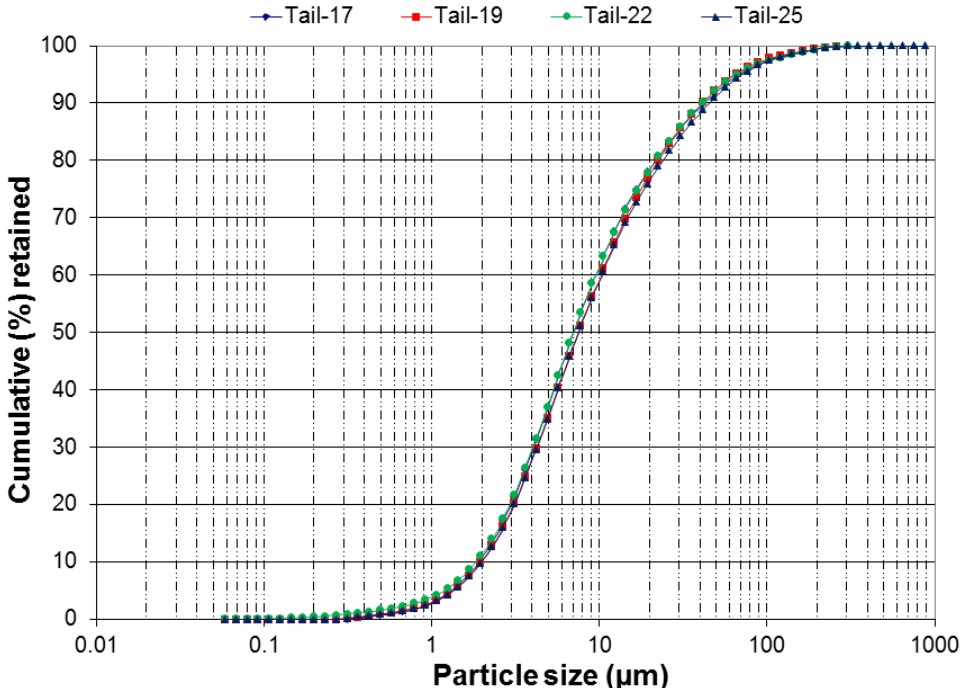

**Figure 4.** Particle size distribution of the four mill tailings samples collected.

The main characteristics of the studied samples are depicted in Table 1. Tail-17 and Tail-19 are characterized by low carbon content (2.74% to 2.87%) compared with Tail-22 and Tail-25. In terms of TOC reactivity, Tail-17 and Tail-22 are far more reactive than Tail-19 and Tail-25. In terms of gold recovery, Tail-17, Tail-19, and Tail-25 present similar grade recovery (91–93%), on the contrary to Tail-22, which is characterized by high carbon and high reactivity (Figure 5).

**Table 1.** Metallurgical characteristics, $C_{tot}$ content, and total organic carbon (TOC) reactivity of samples.

| Sample | Ctot (%) | % Difference [1] | TOC Reactivity | Recovery (%) |
|--------|----------|------------------|----------------|--------------|
| Tail-17 | 2.74 | 41 | Highly reactive | 93 |
| Tail-19 | 2.87 | 32 | Low reactivity | 91 |
| Tail-22 | 3.40 | 45 | Highly reactive | 89 |
| Tail-25 | 3.07 | 27 | Low reactivity | 92 |

[1] % difference = 1 − (recovery with activated carbon/recovery without activated carbon).

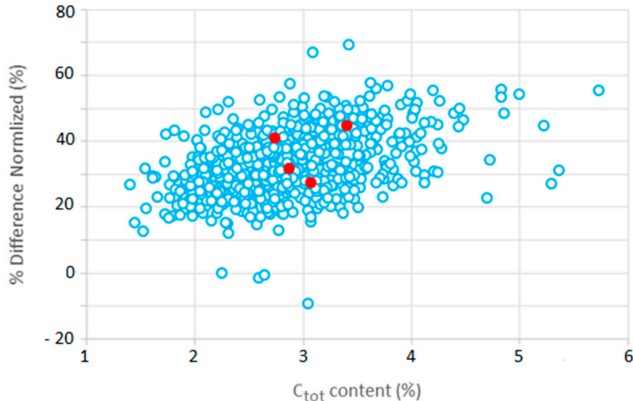

**Figure 5.** Gold recovery as a function of $C_{tot}$ content in studied samples (in red).

The Kittilä CIL tailings have a sulfur content ranging from 4.3 wt. % to 4.8 wt. %. However, a part of the analyzed sulfur is in the sulfate form (e.g., gypsum and anhydrite). Small amounts of carbon are recorded in all samples. Carbon is significantly important in Tail-19, Tail-22, and Tail-25 (2.86–3.38%) compared with Tail-17 (1.83%). Tail-17 and Tail-19 are characterized by significant quantities of gold (2.1–2.23 g/t) in comparison with Tail-22 and Tail-25 (1.63–1.77 g/t). The tailings also contain important contents of silver (11.1–16.5 g/t), iron (14.03%–16.02%), and arsenic (4.3–5.05%). The Si, Al, K, and Na analysis results indicate the presence of silicates and aluminosilicates in the tailings. Note that the sulfur sulfide content is primarily in arsenopyrite ($1.84\% < S_{\text{arsenopyrite}} < 2.04\%$) and pyrite ($2.20\% < S_{\text{pyrite}} < 3.77\%$). Other sulfide minerals (i.e., galena, sphalerite, and chalcopyrite) are present in negligible amounts, as demonstrated by the Pb, Zn, and Cu contents. The results also show that all samples contain small quantities of Sb (400–700 g/t), Ni (90–100 g/t), and Co (15–40 g/t). Fe is the most abundant element in the samples, with a maximum in Tail-25 (16.2%) and a minimum in Tail-22 (14.03%). The chemical compositions of the samples are presented in Table 2.

**Table 2.** Chemical composition of samples Tail-17, Tail-19, Tail-22, and Tail-25.

| Tailings | Au (g/t) | Ag (ppm) | Cu (%) | Zn (%) | Pb (%) | Fe (%) | As (%) | Sb (g/t) | Ni (g/t) | Co (g/t) | $S_{\text{total}}$ (%) |
|---|---|---|---|---|---|---|---|---|---|---|---|
| Tail-17 | 2.1 | 11.1 | 0.025 | 0.011 | 0.007 | 15.18 | 4.78 | 700 | 90 | 30 | 4.28 |
| Tail-19 | 2.23 | 13.6 | 0.026 | 0.021 | 0.007 | 15.77 | 4.3 | 740 | 110 | 15 | 5.67 |
| Tail-22 | 1.77 | 16.5 | 0.027 | 0.015 | 0.005 | 14.03 | 5.05 | 410 | 90 | 30 | 5.68 |
| Tail-25 | 1.63 | 12.20 | 0.028 | 0.021 | 0.008 | 16.2 | 4.64 | 420 | 140 | 40 | 5.80 |

## 3.2. Mineralogical Analyses

### 3.2.1. XRD Results

The mineralogical composition of the tailings determined by XRD is summarized in Figure 6. All samples show similar mineralogical phases, but in different proportions: quartz ($SiO_2$), biotite ($K(Mg,Fe)_3(AlSi_3O_{10}(OH,F)_2)$), gypsum ($CaSO_4 \cdot 2H_2O$), orthoclase ($KAlSi_3O_8$), rutile ($TiO_2$), pyrite ($FeS_2$), jarosite ($KFe_3(SO_4)_2(OH)_6$), muscovite ($KAl_2(Si_3Al)O_{10}(OH,F)_2$), and anhydrite ($CaSO_4$) (Figure 6). Arsenopyrite is not detected by XRD owing to its lower concentration within tailings. Quartz represents more than 24% of the samples, and the remaining phases are mainly silicate and sulfate minerals (i.e., albite, muscovite, jarosite, anhydrite, and gypsum) (Figure 6).

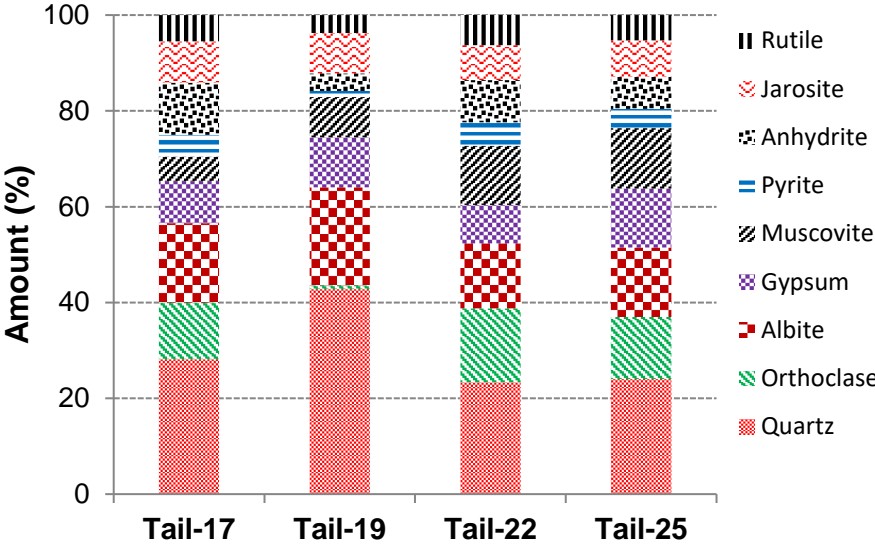

**Figure 6.** Quantification of crystalline phase minerals in samples Tail-17, Tail-19, Tail-22, and Tail-25.

### 3.2.2. SIMS Analysis

One objective of this study was to quantify and establish the distribution of the submicroscopic gold, defined as the atomic gold substituted into the matrix of other minerals, content in the following minerals: pyrite, hematite, and jarosite. The description of the sample analyzed by D-SIMS is provided in Table 3. In total, 100 analyses are performed.

**Table 3.** Gold deportment in the leach tails sample.

| Sample | Leach Tails | |
|---|---|---|
| Gold assayed in the sample | 2.23 g/t | |
| Forms and carriers of gold | Au g/t | % in sample assay |
| Other forms of gold: visible gold, surface gold preg-robbed on carbonaceous matter, soluble gold salts | 0.581 | 26.06 |
| Submicroscopic gold | | |
| Pyrite | 0.260 | 11.68 |
| Hematite | 0.077 | 3.45 |
| Jarosite | 1.312 | 58.81 |
| Subtotal submicroscopic gold | 1.649 | 73.94 |
| Total | 2.23 | 100.00 |

The hematite and jarosite phases very often exhibit complex structures: disseminated in quartz phases, present as intergrowth or forming rims around other particles such as gypsum, rutile, and other gangue minerals. Examples of backscattered electron (BSE) images and SEM/EDX compositional analyses of hematite and jarosite phases are shown in Figure 7.

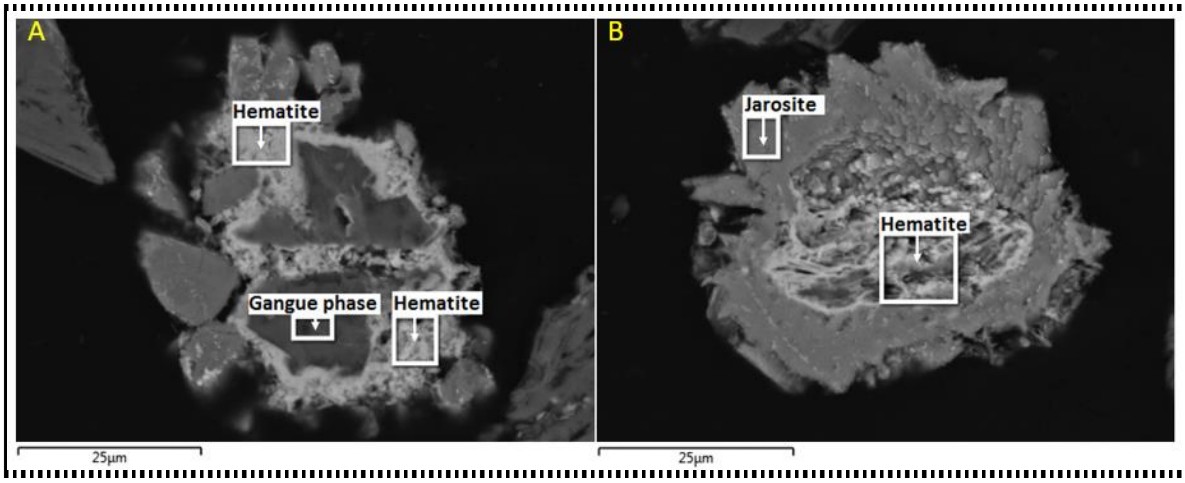

**Figure 7.** Backscattered electron (BSE) image of leach tails sample and energy dispersive X-ray (EDX) compositional analyses of hematite (**A**) and jarosite (**B**).

### 3.2.3. Identified Gold Carriers

The main gold-bearing minerals are pyrite, hematite, and jarosite. Different morphological types of pyrite were identified: coarse (Au 29.09 ppm), porous (Au 11.13 ppm), microcrystalline (Au 15.09 ppm), and disseminated (Au 6.31 ppm) (Figure 8). The pyrite mineral phase is rich in arsenic (As 0.94% to 1.23%; Figure 8). There is a strong positive correlation between the measured Au and As contents in the pyrite mineral phase, as shown in Figure 9. The estimated average gold concentration in the hematite mineral phase is 5.13 ppm, while the estimated average gold concentration in the jarosite mineral phase is 15.43 ppm (Table 3).

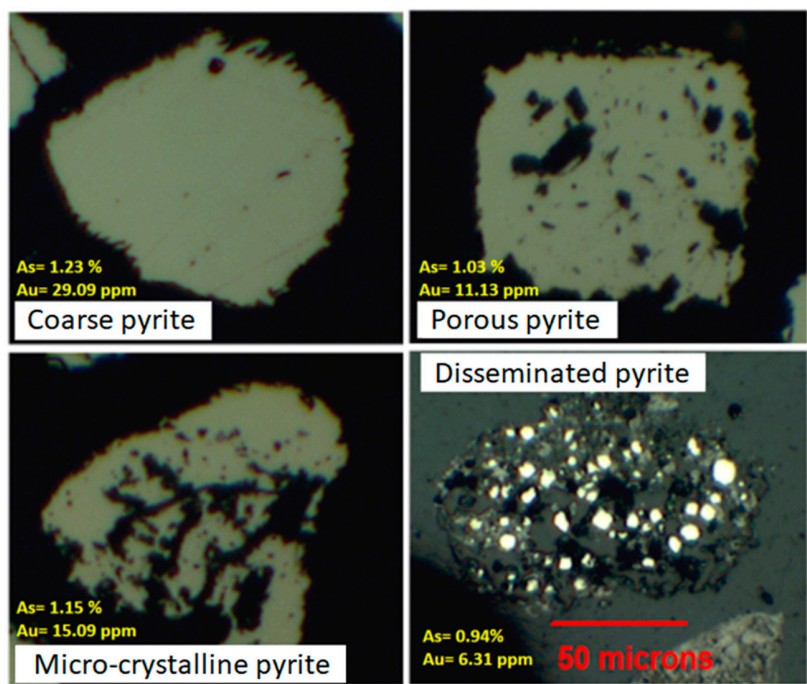

**Figure 8.** Examples of mineral phases analyzed by dynamic secondary ion mass spectrometry (D-SIMS).

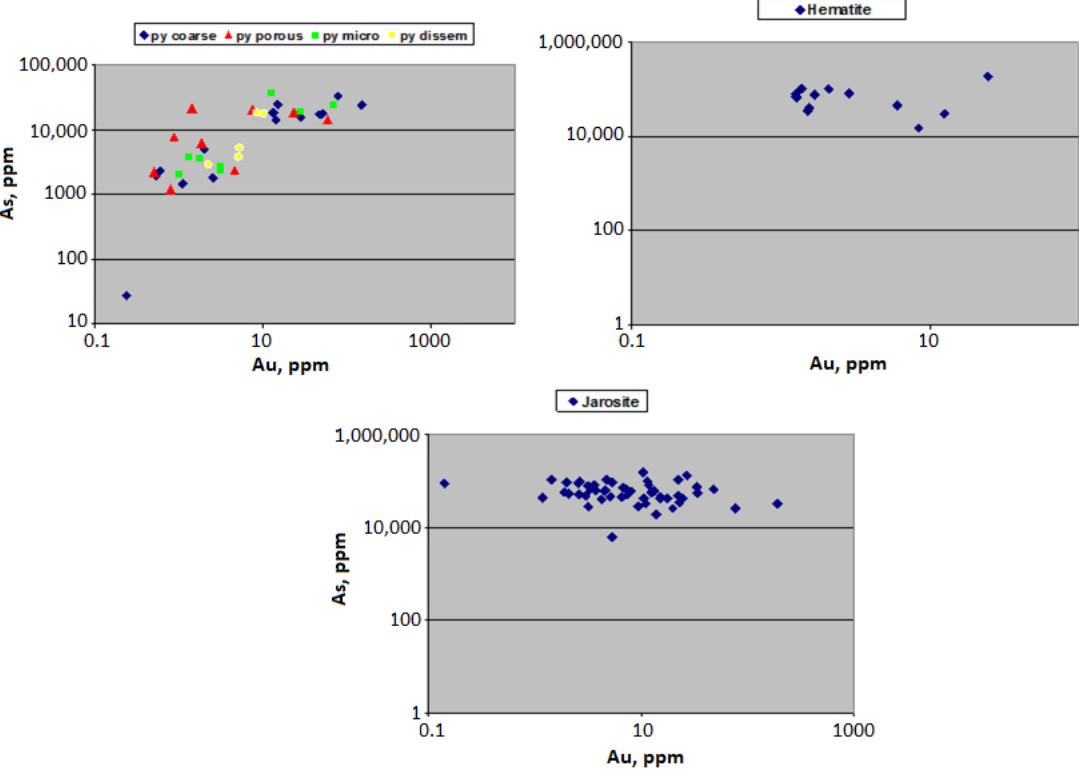

**Figure 9.** Relationship between measured concentrations of submicroscopic gold and arsenic in pyrite, hematite, and jarosite.

### 3.2.4. Submicroscopic Gold Content as Part of the Gold Deportment Balance

The gold deportment balance for the leach tails sample studied is summarized in Table 3 and depicted in the deportment balance diagram in Figure 10. The combined submicroscopic gold contained in the pyrite, hematite, and jarosite mineral phases accounts for 73.9% of the total assayed gold in the

sample. The jarosite mineral phase is the major gold carrier of submicroscopic gold in the sample (58.8%), followed by the pyrite (11.7%) and hematite (3.5%) phases.

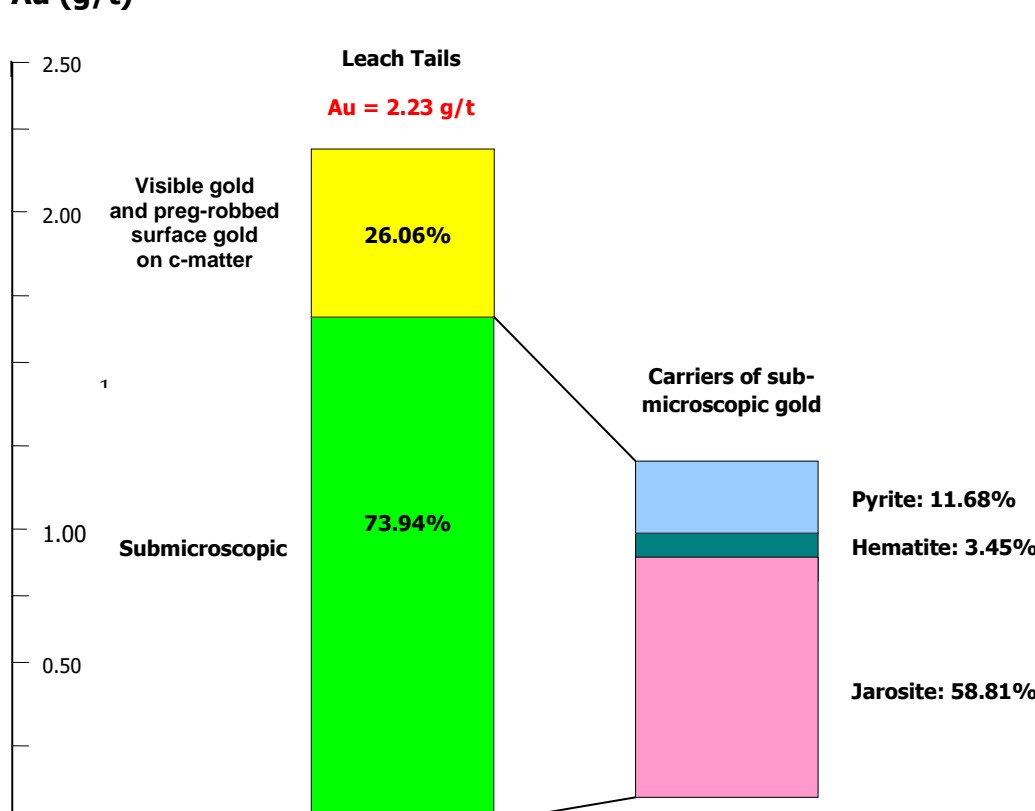

**Figure 10.** Gold deportment diagram, representing all major forms and carriers of gold in the leach tails sample. The relative distribution of gold per carrier is given in % of the total assayed value for gold in the sample.

*3.3. Diagnostic Leach Tests*

Diagnostic leach testing is used to evaluate the distribution of gold in different minerals via a series of selective leaches during leaching phases. The results indicate that about 24.7% of the gold content occurs as free cyanidable gold in all the tailings (Table 4). There was less than 19% free cyanidable gold in Tail-19 and Tail-22. These results (24.7% average gold content) are in coherence with those previously reported in Table 3. An average of 47.4% is locked within digestible minerals (i.e., hematite, jarosite, pyrite, gypsum, and carbonaceous matter). The remaining portion, approximately 27.74%, is locked in gangue minerals (i.e., silicates). Tail-17 and Tail-19 have lower TOC contents compared with Tail-22 and Tail-25. However, the TOC reactivity of Tail-17, Tail-22, and Tail-25 is high in comparison with Tail-19. Tail-19 corresponds to the Kittilä ore with significant gold content, where refractoriness is induced by the dissemination and encapsulation of very fine particles of gold, primarily in oxides, sulphides, carbonaceous matter, and silicates in lesser proportion (Table 4). All the results are summarized in Table 4.

**Table 4.** Results of the diagnostic leach test.

| Description [1] | Gold Distribution (%) | | | |
| --- | --- | --- | --- | --- |
| | Tail-17 | Tail-19 | Tail-22 | Tail-25 |
| Free cyanidable | 28 | 18.9 | 19 | 33 |
| Digestible mineral-locked gold content | 47 | 47.8 | 56.9 | 38 |
| Silicate-locked gold content | 25 | 33 | 24 | 29 |

(1) Only autoclave products (jarosite and hematite) that were analyzed using dynamic secondary ion mass spectrometry (D-SIMS) to confirm diagnostic leach test (DLT) values.

The calculation of the gold distribution within the different minerals was performed using a mass-balance approach. The distribution of total gold content (2 g/t) is shown in Figure 10. The gold distribution exhibits a similar trend in all the samples, except in Tail-25, where the gold content is less than 0.04 g/t within gypsum. In fact, the cyanidable gold varies between 0.32 and 0.56 g/t. The most gold-bearing minerals within the samples are jarosite-trapped (0.23–0.60 g/t), silicates (0.41–0.49 g/t), hematite (0.03–0.09 g/t), and pyrite (0.12–0.26 g/t). However, the gold content associated with carbon does not exceed 0.01 g/t with a maximum in Tail-19 (0.01 g/t). The comparison between the analyzed gold and the calculated gold shows an excellent correlation, with less than a 1.2% error (Figure 11).

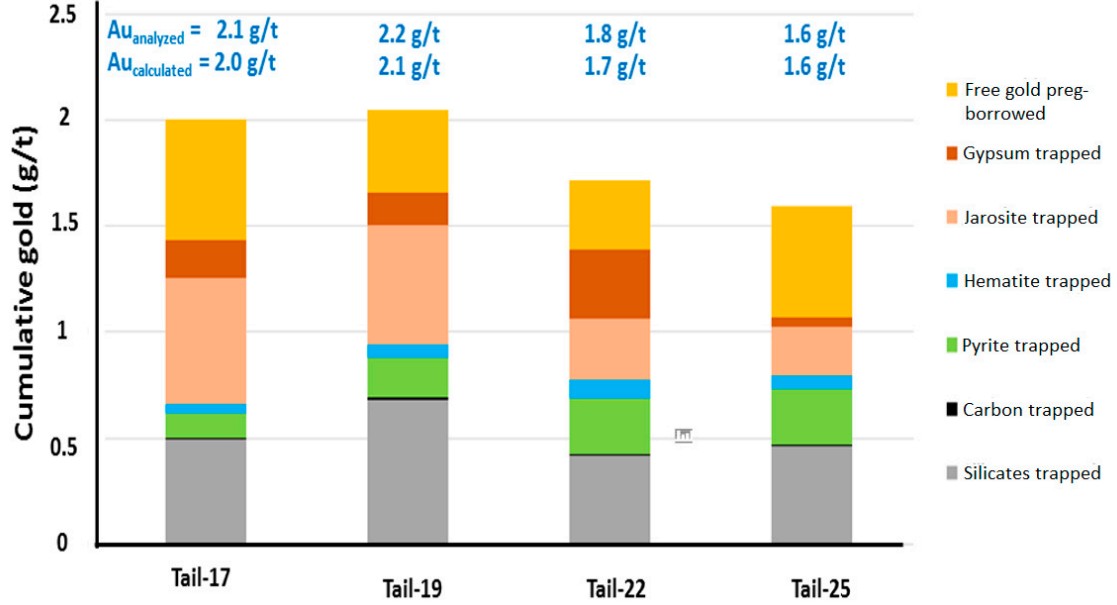

**Figure 11.** Diagnostic leach test results.

*3.4. Adsorption Tests*

The adsorption tests are performed in order to optimize the added gold concentration that it will be used during preg-robbing experiment and to avoid the saturation of the sorption sites during the preg-robbing tests. The purpose of these tests is to assess the efficiency of adsorption of Au in tails. The results obtained using kinetic tests show a linearity of adsorption by C-matter. The lowest doping concentration is selected in order to closely match the real gold concentration in the mill (Figure 12).

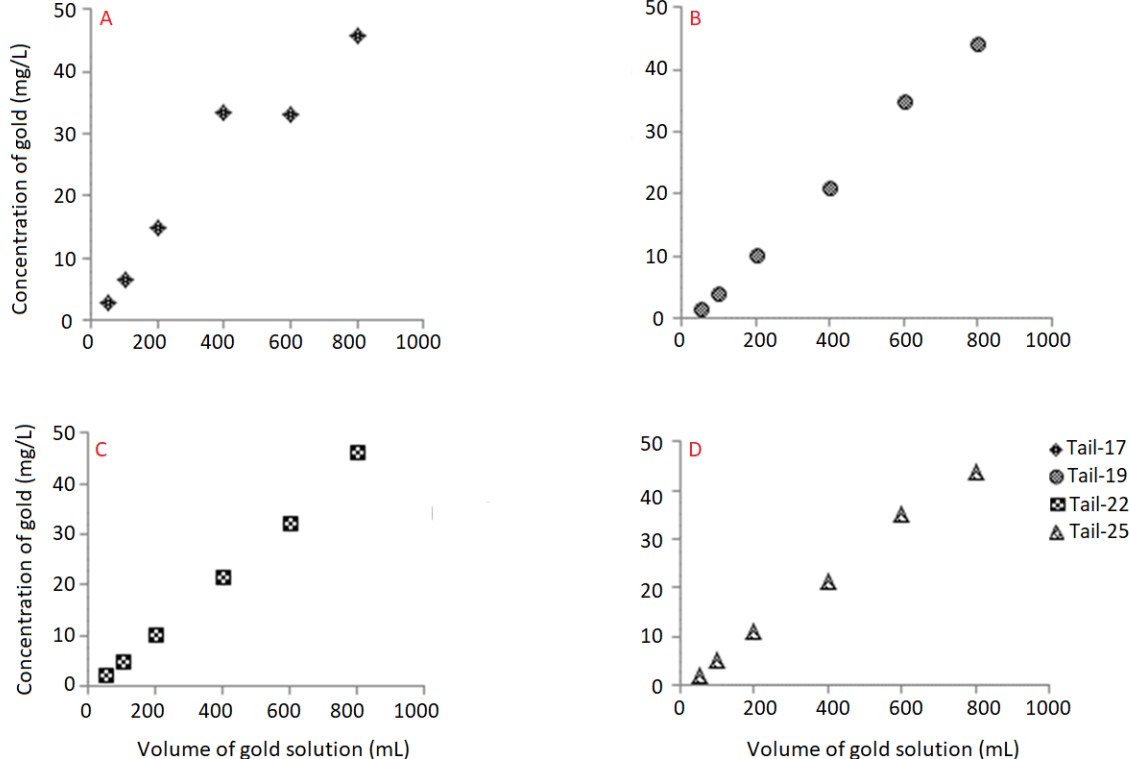

**Figure 12.** Adsorption of gold tests; Tail-17 (**A**), Tail-19 (**B**), Tail-22 (**C**), and Tail-25 (**D**).

The adsorption under a static regime is expressed in terms of percentage (R) of gold removed by adsorbents (tailings) and is calculated as follows:

$$R = \frac{(C_0 - C)}{C_0} * 100 \qquad (1)$$

$C_0$: initial concentration of gold (mg/L);
$C$: residual concentration of gold (mg/L).

The adsorption test of gold on tailings shows that when the initial gold concentration in solution is increased, the percentage adsorbed decreases (Figure 13). An increase in the percentage of adsorption (*R*) as a function of the gold concentration is observed until a maximum value of 20 mg/L, for the four adsorbents. The first point of plateau (saturation point) means that the adsorption sites are saturated; therefore, the total adsorbed gold is relatively constant over the time, despite the initial gold increases. In all samples except Tail-17, the percentage of adsorption (*R*) decreases with the gold concentration until a value of 20 mg/L, and then remains stable at all other concentrations. For Tail-19, Tail-22, and Tail-25, the equilibrium is reached following the addition of 20 mg/L of gold, while for Tail-17, the saturation is achieved after the addition of 60 mg/L of gold. This can be explained by the saturation of the tailings adsorption sites with the increase in gold concentration (Figure 13). In this case, a concentration of 5 mg/L was chosen for the preg-robbing tests.

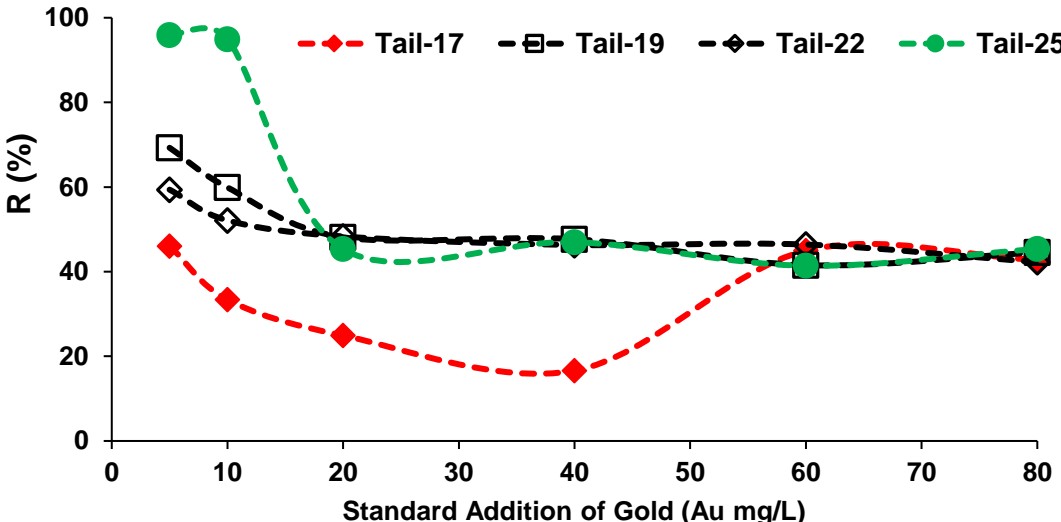

**Figure 13.** Evolution of the gold concentration with the coefficient of percentage (*R*) in the adsorption process at pH = 10.5 and for a tailings mass of 100 g.

### 3.5. Preg-Robbing Tests

The preg-robbing test is conducted using a known amount of standard cyanided gold in contact with samples. After 16 hours of cyanidation, the gold concentration is measured in order to quantify the amount of gold adsorbed by the samples. Indeed, standard additions of gold are used in duplicate for two reasons: to characterize the preg-robbing potential of the studied tailings and to evaluate the repeatability of the laboratory results. The preg-robbing potential of the CIL was highlighted by the strong deviation between two linearities (line with a 1:1 slope, which reflect the absence of adsorption and the trend line of the results). The results show that there is a strong deviation with respect to linearity, which suggests that gold was greatly adsorbed by the samples (Figure 14). It is important to note that all samples present the same behaviour. The addition of 4% (*w/w*) of activated carbon is used to evaluate the strong preg-robbing potential. The results show that all amounts of residual soluble gold are adsorbed by the activated carbon (residual concentration of gold is less than 0.005 mg/L). The four tailings are mostly composed of silicate, iron oxides (hematite), sulfate minerals, and carbonaceous matter. The latter is supposed to exhibit significant preg-robbing potential. The results presented in Table 5 indicate that only preg-borrowing has an important influence on the leaching process of the gold. The PRT values (around 10%) in all samples indicate that the preg-borrowing (PB) phenomenon was dominant compared with the preg-robbing (PR) (Table 5). Further characterization using secondary ion mass spectrometry (SIMS) is needed to identify Au-bearing minerals, both Au-liberated and refractory, and to confirm the results of initial characterization of the leaching behaviour. In this type of experiment, a representative sample in terms of grain size is always required.

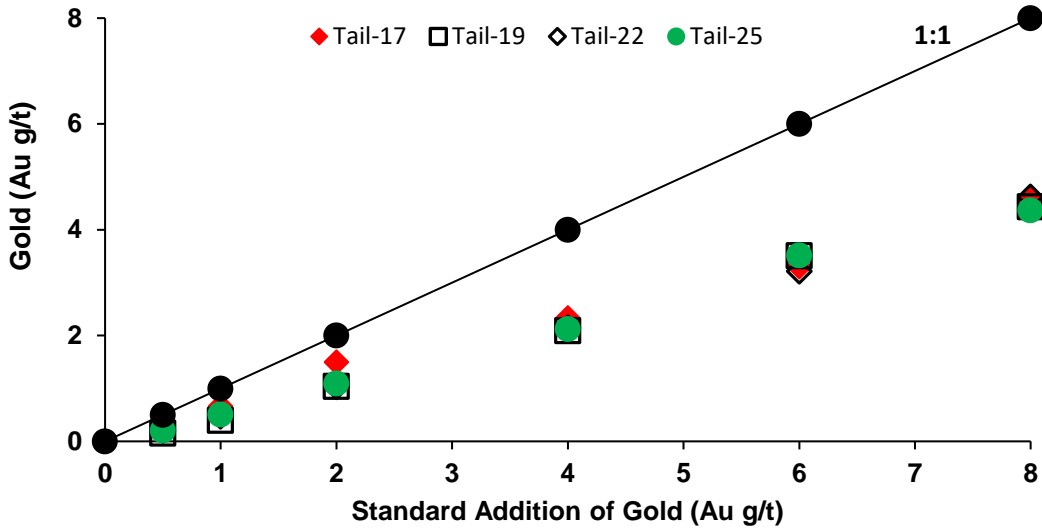

**Figure 14.** Preg-robbing potential of tailings.

**Table 5.** Preg-robbing (PR) tests of studied samples. PB, preg-borrowing; LT, leach tails.

| Parameter | Unit | Tail-17 | Tail-19 | Tail-22 | Tail-25 | Test Conditions |
|---|---|---|---|---|---|---|
| TOC Reactivity | | High | Low | High | Low | |
| Head Gold Grade | g/t | 2.10 | 2.23 | 1.78 | 1.71 | - |
| Tails for Test N | g/t | 2.23 | 2.28 | 1.93 | 1.85 | No activated carbon |
| Tails for Test G | g/t | 12.95 | 12.71 | 10.94 | 10.46 | With gold doping |
| Tails for Test NC | g/t | 1.71 | 1.9 | 1.49 | 1.54 | With activated carbon |
| Tails for Test GC | g/t | 2.84 | 2.92 | 2.58 | 2.49 | With gold doping, and activated carbon |
| **Interpreted Data** | | | | | | |
| PR + PB doped gold | g/t | 10.73 | 10.43 | 9.01 | 8.61 | Test G - N |
| PR doped gold | g/t | 1.14 | 1.03 | 1.09 | 0.95 | Test GC - NC |
| % of PR in PR + PB | % | 11 | 10 | 12 | 11 | PR / (PR + PB) |
| PB in LT | g/t | 0.52 | 0.38 | 0.44 | 0.31 | Test NC - N |
| PR in LT | g/t | 0.06 | 0.04 | 0.06 | 0.04 | Calculated with % of PR |

Analysis of the preg-robbing potential for all samples is summarized in Table 5. The results show a low preg-robbing potential (less than 1% of the residual gold in CIL tailings). This geochemical behaviour is probably caused by a low adsorption and/or precipitation of aurodicyanide within carbonaceous matter. The results obtained for preg-robbing potential are similar to the results found in other sulfide ores where the preg-robbing was driven by sulfide minerals in the presence of free cyanide [41]. In order to achieve good recovery, the refractory fraction of the gold must be leached prior to any cyanidation using a pre-treatment (i.e., bioleaching, pressure oxidation, oxidation roasting) to destroy gold-bearing sulfides.

## 4. Conclusions

The present study was done according to an original methodology that aims to explore, for the first time, the CIL gold loss in oxidized leach tails. It allows to understand the leaching behavior of a double refractory ore using a synergistic approach including mineralogical characterization, diagnostic leach tests, and preg-robbing tests. This procedure is more appropriate for oxidized leach tails because it takes account of the mineralogy of the ore during the procedural development of diagnostic leach tests. The findings are useful to understand the source of the problem (gold losses), and at the same time to optimize the operation for better gold recovery. The free/preg-borrowed gold accounts for 20–35% of the residual gold (0.5 g/t opportunity). The remaining gold content (20–40%; approximately

0.5 g/t opportunity) is trapped within autoclave products (i.e., jarosite, hematite, and gypsum) through couples processes such as the following: (i) gold trapped in hematite is the result of their sorption on the hematite surface and/or their precipitation/coprecipitation with iron; and (ii) gold trapped in jarosite is the result of the geochemical affinity between the gold and arsenic. The results of the mineralogical investigations show similar gold contents in both jarosite and hematite, which suggests that the gold is uniformly dispersed within the analyzed phases. Furthermore, the disseminated pyrite locked in minerals is not attacked under autoclave compared with other morphological types of pyrite, which are partially attacked. All the morphological types of pyrite contain the gold naturally (5–20%) and not preg-robbed; however, the recovery of this gold needs a finer grinding (<50 μm) of the feed materials. Regarding the gold associated with silicates, the authors found that the silicates before the POX contain little or even no gold, but after the autoclave process, the same silicates contain significant gold content (25–35%; approximately 1 g/t), which suggests that this gold is probably trapped by sorption phenomena during autoclave processing. Finally, the preg-robbing results indicate that there is only little irreversible association of gold with carbonaceous matter (<0.1 g/t) = low preg-robbing.

For a better understanding of the geochemical behavior of gold under leaching conditions, further research is needed in the following areas: (i) confirm the occurrence of free/preg-borrowed gold, via mineralogy of gold deportment; (ii) test autoclave operating parameters with SIMS analysis (partial gold deportment), (iii) decrease the remaining sulphide minerals within the tailings and autoclave products by optimizing the energy balance requirements of autoclave, water quality used for the autoclave processing, residence time, pH, Eh, and so on; (iv) conduct partial DLT (cyanidation CYA–HCl–CYA–roasting–CYA); (v) add recyanidation (bottle roll test) of the DLT (longer contact time, more reagent concentrations, or cleaner water); and (vi) confirm silicate locking (grinding, cyanidation, and roasting simulation with gold doped carbon and pure quartz or DLT tails on similar minerals).

**Author Contributions:** Experimental design, M.E. and R.M.-B.; M.E. performed the experiments, analyzed the results, and wrote the paper. Interpretation of results was realized by M.E. under the supervision of R.M.-B., B.P., and M.B.; revision-edition M.E., R.M.-B., B.P., and M.B.

**Funding:** This research was funded by Mitacs (https://www.mitacs.ca/fr) and and Agnico Eagle.

**Acknowledgments:** The authors thank the mining company Kittilä for the great help, especially in the sampling steps; MITACS and Agnico Eagle for the funding support; and Agnico Eagle for the publishing authorization. The authors also sincerely thank UQAT-URSTM for technical support and mineralogical analyses and Western Surface Science for submicroscopic gold analyses.

**Conflicts of Interest:** The authors declare no conflict of interest.

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
