# Peer review of "CIL Gold Loss Characterization within Oxidized Leach Tails: Creating a Synergistic Approach between Mineralogical Characterization, Diagnostic Leach Tests, and Preg-Robbing Tests"

_minerals, doi:10.3390/min9090557_

Round 1
Reviewer 1 Report
Reviewers Comments
General
There are some points that need to be clarified throughout, I’ll provide specific points below.
There is no delineation of the originality of the work. The text discusses that none of the individual techniques are new and the combination of leach and adsorption tests with mineralogy isn’t new either, so the authors should emphasize what is new. Is it the characterisation of this particular material (POx - leach tails)?
Specific
Title
Revise this as you're looking at an oxidised leach tail not an ore.
Abstract
A double refractory ore doesn’t necessarily have solid solution gold, it just has some form of refractory primary refractory issue (gold particles locked in sulphides, solid solution gold in arsenopyrite, refractory gold minerals, etc.) and preg-robbing material. Also, gold is not necessarily associated with carbon in the ore and definitely not in solid solution.
A better definition of the objective of the work would be good (comes back to what is new).
Introduction
Line 40/41: “In refractory ores, the refractory gold is present within the mineral matrix of the Au-bearing minerals.” This is not necessarily the case as with fine particles of gold locked in sulphides and silicates, I assume you’re predominantly meaning submicron gold. Also, it’s a confusing sentence, consider revising.
“Native carbon” is better described as organic carbon.
Line 48/49: Preg-robbing doesn’t occur during flotation, therefore there can’t be any loss due to preg-robbing in the prefloat. Any losses would be due to gold association in the original ore with the prefloat tails.
Lines 58/59: gravity, flotation, roasting, Pox, BioOx – none of these extract gold and the first two aren’t pretreatments, they’re simply benefication to optimise the economics of the process so as not to treat waste.
Lines 61-64: The comment on smelting of other commodities is oddlyplaced in the topic of refractory gold ores.
It would be interesting to discuss the characteristics of the ore before Pox and leaching to show the effectiveness of the oxidation and original leach.
Materials and Methods
Can you define how the TOC reactivity is determined and the purpose of the % difference vs overall recovery is?
Try to avoid providing data obtained in the study in the Materials and Methods section.
Line 146: Examples of D-SIMS are mentioned but then no examples are actually provided for the reader’s reference.
The term “submicroscopic gold” often specifically refers to atomic gold substituted into the matrix of other minerals, define it for the purpose intended in the paper.
Why is lead nitrate used in the leach? This is generally to counteract some mildly refractory nature of an ore induced by sulphides. Is the purpose to mitigate passivation by the remaining sulphides in the tails?
Some more detail around how the Preg-robbing tests were actually conducted and exactly what the N, G, NC, GC and C tests involved would be useful.
Results and discussion
Make sure you refer to all figures and tables in the text.
What is the rest of the composition in addition to what is provided in Table 2? How much Si/Al/K/Ca/etc.? The low proportion of iron and high silica etc. suggests there is a lot of room from improvement in the flotation circuit.
The definition of jarosite as a sulphate is fairly loose, it’s as much an iron hydroxide as it is a sulphate and thus related to hematite. Is it the sodium that’s aiding the formation of the jarosite in the process?
There’s a note that the position of the iron oxide/hydroxyl sulphate is interesting, it is likely that it is positioned in close proximity to the sulphide that was oxidised to form it.
Why is there still easily leachable gold in the leach tails? Should the process residence time/CN concentration/oxygen level/pH in the process be varied to try and recover this gold? If preg-borrowing is the issue then perhaps longer time in contact with the activated carbon?
Line 264: of the diagnostic leach phases, only the nitric acid is an oxidative process.
Line 268: carbonaceous material should only “digestable” if it’s inorganic, if you include roasting in “digestion” then define it as such.
It’s not clear what figure 12 represents, what are they the concentration and volume of?
For the calculation of “R”, wouldn’t it make more sense to be R = (C/C0) *100? Also the discussion of the results in the following paragraph (lines 300-309) is very confusing. The plateau would mean that the residual concentration is proportional to the intial concentration, therefore there is still an increase in total adsorbed gold as the intial gold increases.
Conclusions
The conclusions would also benefit from clarity around the novel outcome of the work. Based on the characterisation I would suggest trying to avoid the formation of jarosite in the autoclave and ensuring leaching to completion to get that ~25 % of easily leachable gold in tails would be the primary targets.
Author Response
Please see the pdf document.

Reviewer 2 Report
The authors have done good work in using different methods to understand the gold losses. The introduction and research work have been described in an interesting and adequate way, but the conclusions part is lacking the wider context. The title says “Creating a Synergistic Approach between Mineralogical Characterization, Diagnostic Leach Tests and Preg-Robbing Tests”. However, the description of this synergistic approach is lacking from the conclusions. The implications of the results in a wider context are missing. The results for Agnico Kittilä mine are inevitably correct, but what do these results mean in a wider context for the scientific community?
Some more specific comments are here below:
Line 19: AEM is mentioned for the first time. Therefore, describe AEM in full(Is it Agnico Eagle Mine?).
Line 52: Some verbs are in the past tense and some in the current. Please check.
Interesting hypothesis were written in the text e.g. in line 83. However, the conclusions around these hypothesis are not clear from the text and should be elaborated.
Line 90-94: In the title of the Figure 1 C and D, the authors write about TOC. In the Figure 1 C and D, Ctot is on the X-axel. Should it be changed to TOC?
Line 97: CIL is mentioned for the first time, so please use Carbon In Leach (CIL).
Lines 62-64 and 128: jarosite, nickel sulfide should be written in small caps.
Line 139: Please open up the abbreviation for D-SIMS.
Table 2: The unit is missing from S total.
Line 235: Open up EDX like was done for BSE.
Figure 12: This figure lacks information and is not self-explanatory. Concentration of what? Volume of what?
Figure 13 title seems to lack something from the beginning.
Table 5 lacks all units and clarifications and needs to be updated.
As a summary, there are various hypothesis in the paper, but the conclusions lack the wider interpretation of the results. The conclusions section needs to be elaborated and answer clearly the research questions and what was promised in the title. Now the experiments have been done in a good and interesting way, but the conclusions section needs further work to combine the results in a wider context.
Author Response
Please see the pdf document.

Round 2
Reviewer 2 Report
My comments have been taken into account and I consider the manuscript to be in an appropriate level for publication.
Author Response
Thank you for the constructive comment.